# Promoting Older Adult Health with Interprofessional Education through Community Based Health Screening

**DOI:** 10.3390/ijerph19116513

**Published:** 2022-05-27

**Authors:** Susan Ostertag, Jade Bosic-Reiniger, Chris Migliaccio, Rachael Zins

**Affiliations:** 1School of Physical Therapy, University of Montana, Missoula, MT 59812, USA; 2Skaggs School of Pharmacy, University of Montana, Missoula, MT 59812, USA; jade.bosic-reiniger@mso.umt.edu (J.B.-R.); rachael.zins@umontana.edu (R.Z.); 3Center for Environmental Health Sciences, University of Montana, Missoula, MT 59812, USA; chris.migliaccio@mso.umt.edu

**Keywords:** interprofessional, teamwork, geriatrics, older adult, health screen

## Abstract

IPHARM (ImProving Health Among Rural Montanans) is a university-based community health screening program that provides valuable interprofessional teamwork and clinical skills training for health care students while addressing the health of older adults. Students perform a variety of health care screenings dependent on the health care professions present and the requests of the community group served. Education, counseling, and recommendations for participants are provided by the interprofessional student teams under supervision and guidance from faculty and clinicians. Supported in part by federal grants such as the Health Service and Resource Administration Geriatric Workforce Enhancement Program (HRSA GWEP), IPHARM has provided interprofessional training for over 2100 students and conducted over 30,000 health screenings at 814 different community events. Surveys from students indicate that the experience promotes effective interprofessional team skills related to communication, an increased understanding of the roles and responsibilities of the health care team, and how to positively impact the health of older adults. These interprofessional screening events for older adults, conducted in the community by health professions students and faculty, help prepare the future workforce for collaborative and effective health care delivery. The purpose of this article is to describe the IPHARM objectives, methods, and impact this program has had on the health of older adults and the training of our future health care workforce.

## 1. Introduction

In 2005 the Montana Geriatric Education Center joined with the Skaggs School of Pharmacy at the University of Montana in operating a health screening program known as IPHARM (ImProving Health Among Rural Montanans) [1]. IPHARM began performing interprofessional health screenings in 2008 for older adults across the state, providing clinical training for interprofessional health care students while striving to improve the health and wellness of older adults. Student, faculty, and clinician involvement over the years has included medicine, pharmacy, nursing, physical therapy, social work, speech pathology, and athletic training. 

IPHARM has helped address limitations that Montana residents have when accessing health care as well as those of the state’s health care providers and students to obtain education and training in geriatrics. With over 145,000 square miles and an estimated population in 2021 of 1,104,271, Montana’s residents are spread over a large area [2]. Older residents, 65 and over, make up 19.3% of the state’s population. By bringing screening events to local communities, the goals for training our future health care workforce and addressing the health care needs of the communities can be addressed. One objective of the Interprofessional Geriatric Health Screenings, performed through IPHARM, is to provide clinical training in geriatrics for health professions students through participation in health screenings for older adults in rural, community-based settings. The second objective is to improve the overall health of older adults in Montana by providing health-related information and education. 

Health care screenings for older adults, conducted in the community by health professions students and faculty, help prepare the future workforce for collaborative and effective health care delivery [3,4,5,6,7]. Several models of this type of work have been reported in the literature, such as providing geriatric screening in an assisted living center [6], developing senior wellness fairs to promote delivery of health care education and screenings while training interprofessional health care students [4], and providing services for low income older or disabled adults in housing units through a wellness initiative [7]. Two additional community-based models were published in 2013, describing interprofessional training in a skilled nursing facility through the University of Washington and the University of Montana geriatric health screening program [5]. It is the latter of these, the health screening program referred to as IPHARM created at the University of Montana, that will be discussed below.

We believe that this model of Interprofessional Education (IPE) demonstrates the effectiveness of community-based health care screenings for both participants and health care students by improving the health of the older adults through conducting health screenings and counseling while helping prepare our future health care workforce to work as an effective member of an interprofessional health care team. Lytvyak et al. found that some health factors can be impacted positively through community-based interventions, though not all, using phone calls as well as physical assessments as part of their multi-center trial [8].

Students participate in health screenings for older adults in rural, community-based settings, often traveling together for hours or days even for just one event, providing opportunities for both formal and informal IPE. An additional focus of these interprofessional events is to provide health-related information and education for older adults based on screening results and personal information obtained through interviewing the participant, which is accomplished by the students and faculty working as an interprofessional team.

## 2. Materials and Methods

The value of interprofessional student training to promote improved health care delivery for older adults is reflected through improvements in the knowledge, skills, and attitude of health professions students due to participating in purposeful and interactive interprofessional education opportunities [9,10]. We are using the definition of IPE from the World Health Organization (WHO) as an experience that “occurs when students from two or more professions learn about, from, and with each other” [11]. The literature supports that interprofessional education (IPE) and learning activities for health care students in a more clinically applicable setting may be more advantageous than provision through didactic or more academic settings by providing a realistic environment for enhancing clinical skills and knowledge, as well as providing an opportunity for reflection and observation [3]. In addition, working directly with older adults as part of an interprofessional student team results in promoting a greater understanding of the roles and responsibilities of other health care professions, enhancing teamwork skills and communication, and promoting a more favorable perception of what working with older adults is like [12].

### 2.1. Planning an Event

Developing safe and effective IPE in any community environment requires a great deal of time, commitment, and human resources [4]. The literature, and our own experiences, have informed us about the complex nature of communities and how the community members themselves are best equipped to determine the most appropriate health screening tests their community members may need or want, as well as what local resources are available for their members. Community members are also more aware of the political landscape of the community and what interventions may be most meaningful [8]. These factors can drive the development of the screening event. For example, finding out from a community member on the planning committee this spring that the bus system does not operate on Saturdays to the local Area Agency on Aging, and thereby limits the ability of people to attend that do not drive, was an important planning consideration with a recent event in Helena MT. Another event in 2021 was influenced by the opposing views related to mask use by the community members during the COVID19 pandemic, which ultimately limited the ability of IPHARM to provide a safe and effective screening event.

Currently at the University of Montana (UM), health care screening events are coordinated by IPHARM faculty and staff at senior community centers, malls, pharmacies, parks, and other venues primarily through networking and making connections with health care providers or students, service clubs, and community groups in local areas. Examples of these community-based connections include Area Agencies on Aging, local hospitals, senior centers, assisted living facilities, health departments, and pharmacies. 

To initiate an event, the IPHARM Coordinator, a Clinical Faculty member at the UM Skaggs School of Pharmacy (SSOP), works with the SSOP Pharmacy Fellow to plan events. A Doctor of Pharmacy student is often available to assist as part of their Advanced Practice Pharmacy Experience (APPE). Other faculty, health care providers, or students from the UM College of Health may also help coordinate IPHARM events. A UM designee will liaison directly with the community member or contact hosting the IPHARM event for scheduling and logistics, including equipment needs, space requirements, and event advertising.

### 2.2. Recruitment and Training of Students and Faculty/Clinicians

Next, an interprofessional group of students is recruited and trained by interprofessional faculty prior to the event. Student participation is dependent on scheduled course and clinical experiences, program requirements for IPE and community outreach, experience, and formal training already completed. Training requirements of students are determined by the supervising faculty and are based on where the student is at in their professional education. Some more advanced students have already received training as part of their didactic course work and laboratory experiences on campus, or through clinical experiences off-campus. Other students are required to complete online training modules directly related to the screening tests and counseling that is provided through IPHARM. A variety of training resources have been developed and maintained by the MGWEP, including online resources and modules that students and faculty can access to ensure safe and effective delivery of health care screening activities and associated counseling, as well as recognizing potential referral needs. The required IPHARM modules include: Screening for Lipid Disorders in Older AdultsFall Prevention in Older AdultsScreening for Osteoporosis in Older AdultsScreening for Diabetes in Older Adults

The pharmacy profession is represented by the IPHARM Coordinator or faculty at these events, with all other health care profession representation being dependent on both faculty and student availability and the event needs. Most commonly, physical therapy and nursing students and faculty will participate in addition to pharmacy students due to student schedules and the requirements of their individual programs. Less commonly, medicine, social work, and athletic training students and faculty or providers have participated. 

While we recognize there is an opportunity for professional overlap in some of the screenings performed, we have found that providing more disciplined, specific training for the screening tests and counseling has had many benefits. Through this process, our students can further develop their own professional identity and skills specific to the screening activity performed. For example, the physical therapy students are trained in the STEADI (Stopping Elderly Accidents, Death, and Injury) Fall Risk Screening and tools promoted by the Centers for Disease Control and Prevention (CDC), which they will use moving forward as Doctor of Physical Therapy practitioners [13]. In addition, maintaining expertise with the screening tests allows the students to provide more in-depth education and training for the community participants and their student team colleagues from other professions. The students can also demonstrate a more effective and efficient performance of the screening tests at the event, resulting in a safer and more satisfying experience for both the students and the participants. 

### 2.3. Activities Day of the Event

The event itself is the highlight for everyone. Consistent with other authors, the time commitment must include several hours outside of the scheduled event itself for set-up, clean up, connecting with local community leaders, as well as training and debriefing with students and clinicians [4,6]. Faculty and local clinicians are recruited and trained by IPHARM to assist with the supervision and mentoring of students at the event site. Under supervision, students perform a variety of health care screenings, most commonly including lipid panels, fall risk, medication reviews, bone density, diabetes management, and vital signs. Screening activities are determined by several factors:Grant funding requirements;

For example, current funding through the Montana Geriatric Workforce Enhancement Program (MGWEP) requires several specific health screenings be performed. Faculty and students are trained by representatives from MGWEP to ensure that data reporting requirements are met.

2.Faculty and student availability;

An illustration of this occurred at an event performed at the local county library, including education and screening activities for mild head injury and concussions due to the interest and opportunity provided by UM Athletic Training students.

3.Community member needs or requests;

An example of an activity outside the traditional grant-related screenings includes a request to screen for lower extremity edema, triggered by a desire of local providers to promote access to local physical therapists and provide education regarding the role that Physical Therapy has in managing lymphedema.

4.Implementing best practice and evidence;

Screening grip strength and/or cognition is an example of how implementing other tests reflects the incorporation of best practice [14].

Education, counseling, and any recommendations for follow-up or referrals for participants are provided by the interprofessional student teams under supervision and guidance from faculty and clinicians once the screenings are completed. 

Interprofessional student teams are coordinated by the supervising faculty and clinicians at the event. The size and profession mix of these teams is based on the number of students from each profession attending and varies between events. Thompson et al. demonstrated that an equal number of professional representation is not necessary to promote IPE among health care students [10]. The objectives for working as part of an interprofessional team at the event are shared with the students after introductions are completed. The students are oriented to the expected flow of participants, informed consent requirements, and the various screening tests to be performed. In this way, we create diverse interprofessional student teams that can explore and learn about the roles and responsibilities of other team members while they work on the development and practice of effective communication with each other and their community participants. This method of promoting IPE is consistent with Ezzendine and Price, who found that forming student teams, facilitating student introductions, and facilitating team dynamics were valuable [15].

### 2.4. Assessments of the Event and Training

The final piece of these IPHARM activities is to assess the event’s effectiveness and training. Assessments include both in-person and survey methods. Immediately following an event, supervising faculty will huddle with students to verbally review the student’s performance, and any questions students have about the participant results. Survey assessments are completed by the students and the faculty or clinicians within one-week post-event, and the participants are asked to complete surveys via paper or tablet at the event site once the screenings are completed. Surveys from students indicate overwhelmingly that the experience promotes effective interprofessional team skills related to communication, increased understanding of the roles and responsibilities of the health care team, and how to positively impact the health of older adults. One student commented on the evaluation form that “It was interesting to witness the collaboration between PTs and Pharm students to provide the best possible care—this highlighted each professions specialty in terms of how to assess for fall risk and in offering preventive health care education.” Information about these assessments is below in “results”. 

## 3. Results

IPHARM has provided interprofessional training for over 2100 students and conducted over 30,000 health screenings at 814 different community events across Montana. The number of screenings performed per Montana county is indicated in Figure 1. With more detailed data collected from 2012–2020 as part of grant funding requirements, we screened 5197 adults 55 and older.

### 3.1. Student Evaluations

From 2017 to 2020, IPHARM collected health screening evaluations from students who participated in events through an online survey program (Survey Monkey) before moving to another survey system (Qualtrics) in 2021. During this period, responses were collected from 138 students from disciplines including pharmacy, nursing, physical therapy, medicine, and social work (see Table 1).

99% of responders (n = 137) indicated they were adequately trained for the event, and over 98% (n = 136) indicated that supervision was helpful towards clinical education.Students shared what they learned from other health professions during this period. One student stated, “It was helpful to see how the other students went about completing their assessments and comparing strategies they used to my own.” Another student said they learned “roles of other disciplines, how our interactions with participants can overlap and mesh together.” Students also learned “how to handle patients with various needs at the time of assessment (unsteady on their feet, hard of hearing, etc.)”.Over 98% of students (n = 136) felt they had a positive impact on participants’ health education.

Table 2 demonstrates the number of events and students, the number of older adults screened, and the number of screening tests performed from 2017–2021. This timeline was chosen as it provides several years of data and a clear representation of the impact IPHARM had up until the COVID19 pandemic arrived in Montana in March of 2020.

Table 3 provides a breakdown of the students’ professions and participation by year between 2017–2021.

### 3.2. Faculty/Clinician Assessments of Student Performance

Faculty/Clinician preceptors are present at each health screening event to help facilitate student learning and guide patient interaction. From 2017 to 2021, at least thirteen individual preceptors from four different disciplines participated in 138 events. 

Early evaluations in the program included faculty evaluations of students’ professionalism and ability to properly conduct the point-of-care test. Questions included: Did the trainee introduce themselves to the client?Did the trainee briefly explain the procedure prior to beginning?Did the trainee exhibit understanding of the instrument being used?Did the trainee properly counsel the client about their results?

Students were required to “pass” the clinician/faculty member’s evaluation before they could work alone with less supervision. Assessments performed by supervising clinicians and faculty showed growth of pharmacy students over a span of events which usually required an average of three events to achieve proficiency. 

The student performance assessment was modified in 2020, and a seven-question survey was initiated to better gather information about skills related to IPE, such as communication with other students. On a five-point scale from “strongly agree” to “strongly disagree,” students are assessed on their ability to

Demonstrate flexibility during a health screening eventDemonstrate professionalism in manner, communication, appearance, and practiceEstablish good patient rapportGather relevant subjective and objective information from the patientEffectively communicate (verbally and non-verbally) with the patientEffectively communicate (verbally and non-verbally) with other students and preceptorsDemonstrate understanding of devices, questionnaire, or tools used

When health screening events commenced again in 2021, eleven students were evaluated. All eleven students scored “somewhat agree” or “strongly agree” in each category, indicating that the interprofessional health screening events effectively promoted student skillsets related to both IPE and their specific professional activities. 

### 3.3. Participant Assessment

Postage paid notecards are given to participants that received abnormal results at the health screening event. Participants are asked to return the cards within three months, indicating if they had made lifestyle modifications, followed up with their provider, started a medication, quit smoking, or made no changes. They are allowed to select multiple answers. Participants are also asked how satisfied they were with the health screening visit and if they gained a better understanding of their health due to this screening service. 

From 2003 to 2007, 286 outcome cards were returned to IPHARM, but there is no record of how many were sent or given to participants. Data collection methods were modified to better track participant responses in 2008. As a result of this change, we know that 3488 outcome cards were distributed from 2008 to March 2020, with 2263 returned, demonstrating a 65% return rate. Of the responders, 2000 (88%) indicated they had taken at least one action to improve their health. Over 2200 participants filled out the satisfaction survey, with 2112 (96%) indicating they were satisfied or very satisfied with their IPHARM visit. Over 1700 participants completed surveys on patient understanding, and 1535 (90%) agreed or strongly agreed that they gained a better understanding of their health due to this screening service. 

The COVID19 pandemic resulted in the cancellation of in-person screening events beginning in March of 2020. To continue providing this service to community members and train our future health care workers, IPHARM developed an online health screening format. Screenings were performed through the completion of a questionnaire that was emailed to the participant ahead of time, as well as through synchronous questions and discussions with the participants using telehealth. One example of a series of questions asked is from the “Stay Independent Brochure”, an evidence-based screening measure for identifying fall risk from the CDC. This brochure is part of a training and education program referred to as STEADI [13]. Despite the low number of participants (7), we trained 24 health care students representing 4 different professions. This low number of participants was not surprising given the challenges older adults have accessing and using telehealth [16]. With the lifting of pandemic restrictions, IPHARM is once again planning and carrying out in-person community-based events. 

## 4. Discussion

Initially, IPHARM was developed to help meet the needs of rural Montana residents with a team of pharmacists and their students [1]. Since then, interprofessional education and teamwork skills have been incorporated into the procedures and practices while continuing to serve Montana residents. More recently, in response to faculty and student input, as well as published literature and faculty development, we have made positive changes to our processes [4]. First of all, IPE has been enhanced through a more structured grouping of students as the student health care team. When possible, and based on student-to-participant ratios, this interprofessional team escorts one older adult through various screening stations, with each student counseling and educating participants as appropriate in the team setting. In addition, students will also “test” each other, providing opportunities for enhanced communication skills and experiential learning about various roles and responsibilities within each profession. Additional adjustments have been necessitated by the COVID-19 pandemic, including the implementation of telehealth strategies for participant screenings, registering for screening events via online links and phone calls, as well as the implementation of personal protective equipment and social distancing.

Lessons we have learned along the way are many, and have also been reported by other authors:Providing direction and structure for interprofessional teamwork skills training is necessary, as health care students tend to congregate within their profession [6].Managing expectations of the students is important as students at the same event or volunteering for different events may not have the same experience [4]. The flow of participants may vary dependent on the time of day or competing activities. Additionally, weather, location, parking availability, and marketing efforts all may impact the number of participants attending, and, ultimately, the experience of the students.Community members and groups are valuable partners in the planning process for these events [4]. They understand the potential partners, referral resources, programs, and logistics of their community better than anyone else. Examples: Bus routes and schedules to ensure that participants can attend; daily schedules and events such as Bingo at the Senior Center; what resources are available in the community to refer participants to.Recruitment of students may be challenging and can be strongly influenced by academic support by incorporating assignments related to volunteering, providing academic credit for participating, and arranging class times to accommodate the scheduling of interprofessional activities [4,6].

We would also like to share the fact that there were several challenges pulling the data, outcomes, and survey results for this publication. Over the past 20+ years, IPHARM has had a number of different databases, advocates, coordinators, and staff members setting up, reporting, and managing the results of the health screenings performed. Per Telari and Goyle, missing data and other limitations exist with retrospective studies as they rely on the accuracy and thoroughness, as well as organization, of others [17]. As a result, we have chosen select time periods that we considered best represented IPHARM through the activities and outcomes achieved.

## 5. Conclusions

Over the past 20 years, the IPHARM adult health screening program at the University of Montana has made an impact on the state of Montana by providing health screenings and promoting improved health behaviors, as well as through the training of our future health care workforce with intentional interprofessional team skills training and clinical skills development. This article provided an opportunity for us to describe the methods and outcomes of our adult health screening program at the University of Montana, as well as provide information about the impact that these community screens have had on the health of older adults and the training of our future health care workforce. What started as a one-year funded grant opportunity operating out of an upgraded 1991 motorhome transitioned into IPHARM with funding through the Montana Geriatric Education Center. Funding since then has been provided by HRSA through the MGWEP primarily. The activities, goals, and processes for IPHARM have continually adapted to meet the needs of students, faculty, funding agencies, and communities. With support in the literature for IPE outside of the classroom environment that is immersive and impactful, IPHARM is well-equipped to provide support and services for older adults and better prepare our future health care workforce.

## Figures and Tables

**Figure 1 ijerph-19-06513-f001:**
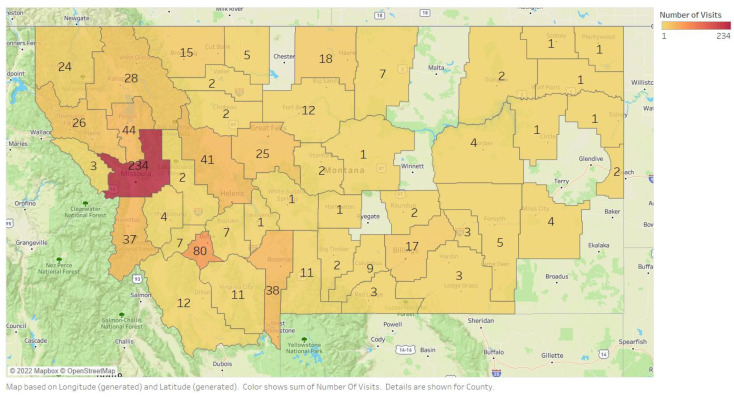
The numbers in each county represent how many IPHARM events occurredfrom 2003–2021 per county across Montana.

**Table 1 ijerph-19-06513-t001:** Number of student participants that were sent an evaluation and number of responses from 2017–2021.

Calendar Year	Students	Responses
2017	238	24
2018	210	35
2019	188	33
2020	29	24
2021	26	22
Total	691	138

**Table 2 ijerph-19-06513-t002:** Number of events, students, older adults screened, and screening tests performed from 2017–2021.

Calendar Year	Events	Students	Screened 55+ Years Old	Tests Performed
2017	51	238	695	1263
2018	42	210	651	1117
2019	30	188	545	1222
2020	7	29	37	100
2021	8	26	53	131
Total	138	691	1981	3833

**Table 3 ijerph-19-06513-t003:** Number of students’ professions and participation by year between 2017–2021.

Calendar Year	Pharmacy	Physical Therapy	Nursing	Social Work	Medical	Other	Total
2017	166	28	41	3	0	0	238
2018	130	39	18	22	1	0	210
2019	93	64	26	4	0	1	188
2020	10	16	3	0	0	0	29
2021	8	8	7	0	0	3	26
Total	407	155	95	29	1	4	691

## Data Availability

Not applicable.

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
