# Peer review of "Promoting Older Adult Health with Interprofessional Education through Community Based Health Screening"

_ijerph, 2022, doi:10.3390/ijerph19116513_

Round 1
Reviewer 1 Report
Thank you very much for allowing me to evaluate your paper.
Unfortunately, I think it has important shortcomings.
Already in the title it is not clear what the aim of the research is. It is not clear whether the aim is to evaluate the satisfaction of individuals or students.
The abstract is unstructured and difficult to understand.
There is no exact definition of IPE in the introduction.
Older people are considered to be 55 years of age or older, but this choice is not justified.
The introduction does not provide all the necessary data to get an idea of the framework of the study.
There is also no clear objective in the introduction.
The methodology only describes the programme carried out and in a general way. Important flaws are detected such as the fact that nursing, according to figure 1, is not the one who participates the most together with physiotherapy. It does not appear that the number of students required for each event is programmed, nor is it specified what the functions of each professional category are. It is not clear how students are trained, especially first-time trainees. Only one instrument is known to be used by physiotherapists. It is not known how bone density is measured and which vital signs are evaluated. It is not clear whether all measurements are taken by all students and whether they are used to replace each other's functions. In short, the programme does not seem to have a clear plan but is defined as needs arise.
There is talk of assessing learners verbally at each event, but it is not clear whether this is done formally or informally and how this information is studied. It is also stated that students are evaluated by means of a survey, but it is not known whether it is a satisfaction survey, an ad hoc survey or a response rate. Nor is the sample size specified. This part would really be the methodology and the previous part should appear in the introduction.
Figure 1 is named as 2 and furthermore does not contribute anything to the study.
Figure 2 should be illustration 1 and should be cited in the text.
Teacher evaluation is not mentioned in the methodology, but it is mentioned in the results. They do not specify what criteria they use to determine that a learner is competent, nor what competences are needed. Firstly, they use a YES/NO questionnaire and then a Likert scale. The two scales are not comparable.
Again, an assessment is made which was not previously mentioned in the methodology. This time it is the assessment of the patient. It is not known at what point they are assessed and whether they are all assessed at the same time.
When telehealth is used, what measurements have you carried out?
The dissuasion does not compare the results with other authors.
The limitations are many and are admitted by the team.
The conclusions do not answer the objective because the objective is not clear.
To publish these data, it is necessary to have the approval of a research ethics committee.
The number of bibliographical references is very low.
I hope to have contributed to the improvement of the paper.
Reviewer 2 Report
IPHARM (ImProving Health Among Rural Montanans) is a unique interprofessional ed-11 ucation (IPE) effort that provides valuable team work and clinical skills training for health care 12 students while addressing the health of older adults. This article is very applicable, and long-term follow-up.
This article is well written, but some minor issues need to be fixed:
1. Providing direction and structure for interprofessional teamwork skills training is 247 necessary, as health care students tend to congregate within their own profession. I suggested to add more discussion and related literature to this topic in this paragraph.
2. Community members and groups are valuable partners in the planning process for 254 these events.[4] They best understand the potential partners, referral resources, pro-255 grams, and logistics of their community better than anyone else. Community cooperation is an important topic, and it is recommended to increase discussion and related literature citations in this area.
Reviewer 3 Report
This is a descriptive assessment of an interdisciplinary training program for undergraduate (or pre-clinical) health science students over 10-year period 2012 to 2021 in a rural US state that if focused on ensuring that these students have exposure to and experience with older adults in clinical encounters by conducting health screenings in a wide variety of community settings across a considerable geographic area. This is part of the GWEP national initiative and a few sentences about the larger context might help readers understand this.
Because this is a largely descriptive study, the methodology – using both self reported student and faculty assessments of the experience, as well as some data from elders who were being screened are clearly important-- but equally clearly they are not exhaustive—and that’s fine. However, the manuscript seems to say that the methodology using different data sources evolved over time with the student assessments by faculty formalized into a series of 8 questions requiring explicit judgements by faculty preceptors on a 5 point ordinal scale. However no results from that particular faculty 8-item faculty assessment are presented. Hence, discussion of that particular scaling method seem redundant and can cause confusion for a reader--unless some actual quantitative data from those relatively few, but perhaps important, cases are actual
The student performance assessment methodology is described but no data are presented for the 2020-2021 period in the table. . Understandably this is the apex of the pandemic, but it remains unclear how many students were enrolled in each year over the decade long life of the program.
These are really large numbers, especially from a sparsely populated state with seniors who have health care, transportation, and housing challenges: 2012-2021---814 events over 2,100 students; 13,671 screenings with data from 5197 adults. Despite the fact that the actual data the authors seem to have in hand are from much smaller numbers of more recent cohorts of students and elders, the
It would be helpful to have a clear denominator for each of the relevant population groups specifically described for the period 2017 through 2019 (or even through 2021 if possible) where most of the discussion is focused and most of the quantitative data seem to have been collected or compiled .
Ideally the authors should pick a time period where they have a reasonable amount of data and specify each the following quantities for that time period:
(a) the number of training events taking place;
(b) the number of students involved in those training events (whether or not data were obtained from those students) ;
(c) the number of students involved in those training events from which useable data were obtained;
(d) the number of faculty involved in these events; training (whether or not data were obtained from those faculty);
(e) the number of students involved in those training events from which useable data were obtained whether or not data were obtained from those students) ;
(f) the number of elders involved in these events; (whether or not data were obtained from those elders);
(g) the number of elders involved in those training events from which useable data were obtained.
Having all these quantities will allow readers to make sense of how representative any of the data obtained (e.g., a response rate) is for assessing in some meaningful way the potential impact of the program as a whole on the students, the faculty, and the community participants
Then at lease some of quantitative measures obtained from each of the three groups: student self assessments, faculty assessments of student performance, and community elder participants assessment of the experience—should be succinctly presented in tabular or graphic format.
You seem to have something more important here than you realize even if your quantitative database is relatively small and sparse.
Round 2
Reviewer 1 Report
Dear authors,
congratulations for the improvements in the paper. I think they have made a great effort to adapt it to the proposed suggestions.
My most sincere congratulations.
King regards.
Reviewer 3 Report
I believe that the article does has not addressed all the issues raised in my initial review. Hopefully the presentation of the quantitative data might strengthen the paper and can be accomplished within the publication scheduling constraints.
Round 3
Reviewer 3 Report
Recommendations that this reviewer made have been adequately addressed